# TA-MoE: Topology-Aware Large Scale Mixture-of-Expert Training

**Chang Chen[1]\*[†] Min Li[2,3]\*, Zhihua Wu[4], Dianhai Yu[4], Chao Yang[2,3,5][‡]**
[1]Center for Data Science, Peking University
[2]School of Mathematics Sciences, Peking University
[3]National Engineering Laboratory for Big Data Analysis and Applications, Peking University
[4]Baidu Inc.
[5]Institute for Computing and Digital Economy, Peking University
`charlie_chen,chao_yang@pku.edu.cn`
`limin_cn@163.com`
`wuzhihua02,yudianhai@baidu.com`

## Abstract

Sparsely gated Mixture-of-Expert (MoE) has demonstrated its effectiveness in scaling up deep neural networks to an extreme scale. Despite that numerous efforts have been made to improve the performance of MoE from the model design or system optimization perspective, existing MoE dispatch patterns are still not able to fully exploit the underlying heterogeneous network environments. In this paper, we propose TA-MoE, a topology-aware routing strategy for large-scale MoE trainging, from a model-system co-design perspective, which can dynamically adjust the MoE dispatch pattern according to the network topology. Based on communication modeling, we abstract the dispatch problem into an optimization objective and obtain the approximate dispatch pattern under different topologies. On top of that, we design a topology-aware auxiliary loss, which can adaptively route the data to fit in the underlying topology without sacrificing the model accuracy. Experiments show that TA-MoE can substantially outperform its counterparts on various hardware and model configurations, with roughly 1.01x-1.61x, 1.01x-4.77x, 1.25x-1.54x improvements over the popular DeepSpeed-MoE, FastMoE and FasterMoE systems.

## 1 Introduction

The scale of model parameters in neural networks has increased from millions to trillions in recent years, which promotes model accuracy in many domain, such as language processing [3, 4, 5] and computer vision [27, 23]. However, the limited hardware resources, e.g., memory capability and communication bandwidth, have constrained the model size to further scale up. To relieve this tension and improve the model performance, Mixture of Expert (MoE) with a sparsely gated structure was recently reintroduced [16, 26, 25]. The core structure of MoE is a group of small "expert" networks and a gate network. Guided by the gate result, input data is dynamically routed to only a sub-group of experts for computation. Compared with dense training methods, the sparsely activated feature of MoE can significantly reduce the computation burden, extend the model size, and achieve higher accuracy [6, 7, 11, 13].

---

\*Equal Contribution.
[†]Work done during internship at Baidu Inc..
[‡]Corresponding author.

36th Conference on Neural Information Processing Systems (NeurIPS 2022).

Since MoE plays a vital role in large-scale model training, efficient MoE parallel training has recently received much attention. As one of the most popular MoE training approaches (Figure 1), expert parallelism [11, 7] distributes experts to different devices, and each device is responsible for a different batch of training samples. Correspondingly, extra global communication is necessary for data exchanges among devices. Recent works aim to increase expert parallelism performance from two aspects. On the one hand, the dynamic pattern of MoE results in severe computation load-imbalance problems that a small number of experts may receive, process, and send the majority of data. Several approaches were proposed to make full use of the available experts, such as adding an auxiliary loss [26], controlling expert capacity [11, 7], and optimizing the assignment scheme for a balanced load [12, 28, 22]. On the other hand, global communication is another main obstacle to efficient MoE training. Most of the recent works reduced the communication cost from a system perspective, such as computation and communication overlapping [9], customized communication operation acceleration [21, 18], and adaptive routing [17].

In addition to the continuing efforts made to improve the performance of MoE, there are still two major challenges. With the development of the complicated distributed network environments, the existing even dispatch method may cause network contention in the slowest links, leading to poor communication performance, especially on heterogeneous networks. Although a few early works [9] have proposed methods to dispatch more data to slow links, these methods may make the expert load imbalanced and could influence the model accuracy. Efficient communication demands more delicate dispatch strategies. How to improve the training efficiency without sacrificing the model accuracy is still worth studying. Besides, most of the existing communication optimizations for MoE [21, 17] are studied with a specific hardware environment. How to develop methods that can adapt to a variety of hardware environments is also of great practical value.

To tackle these challenges, we design TA-MoE, a topology-aware large scale MoE training method that can adaptively adjust the communication volume to fit the underlying network topology. By abstracting the dispatch problem into an optimization objective based on the communication modeling, we obtain the approximate dispatch pattern under different topologies. On top of that, an auxiliary topology loss with pattern-related coefficients is proposed, which can dynamically adjust the communication volume without interfering with the model convergence. TA-MoE can also be easily incorporated into the widely used MoE systems, such as DeepSpeed-MoE [21] and FastMoE [8].

We conduct experiments on various typical network topologies and model configurations. Results show that TA-MoE can substantially outperform DeepSpeed-MoE and FastMoE with roughly 1.01x-1.61x speedup and 1.01x-4.77x speedup on different configurations without sacrificing the model accuracy. Compared with the recently proposed Hir gate of FasterMoE, our method can achieve 1.25x-1.54x speedup on time to convergence. Besides, a more detailed analysis of communication and data dispatch pattern further demonstrates the effectiveness of the proposed data dispatch strategy. The code of TA-MoE is available at: https://github.com/Chen-Chang/TA-MoE

## 2 Related Work

Several frameworks have featured sophisticated designs to support efficient MoE training. GShard [11] and DeepSpeed-MoE [21] subtly composed several einsum operators into the computation of MoE but introduced redundant zero computation and extra memory consumption. FastMoE [8] customized the essential computation kernels to improve resource utilization effectively. To further enhance the performance, most of the systems adopted an auxiliary loss [26] to achieve an even dispatch pattern and enforced the number of data processed by each expert below some uniform capacity. Based on these popular implementations, recent works aim to improve the MoE training performance from mainly two aspects: model structure design and communication optimization.

From the perspective of model design, BASE Layer [12] and the work of expert choice routing [28] assigned an equal number of tokens to each expert by delicate designs of the gate. Instead of learning the weight of the gate, Hash Layers [22] adopted an efficient hash function to guide the dispatch. The hybrid structure of PR-MoE [21] improved the parameter efficiency by fixing one shared expert. BaGuaLu [15] re-distributed the data chunks evenly, damaging the model accuracy. However, almost all of these high-level algorithms are agnostic of the complicated underlying hardware effect on training performance.

As for communication optimization, DeepSpeed-MoE [21] and HetuMoe [18] implemented a hierarchical all-to-all communication kernel to improve network utilization. Tutle [17] designed adaptive routing techniques coupled with a specific network architecture. Despite of these delicate designs, the improvement space of system-level optimization is significantly constrained by the dispatch patterns of MoE. Recently, FasterMoE [9] made an initial try to take the dispatch pattern into consideration by setting a compulsory ratio of intra-node to inter-node dispatch chunk sizes but sacrificed some model accuracy. In this paper, we propose a topology-aware routing strategy that enables an efficient communication pattern to fit into the underlying topology without sacrificing the convergence performance.

## 3 Background

### 3.1 MoE Model Structure

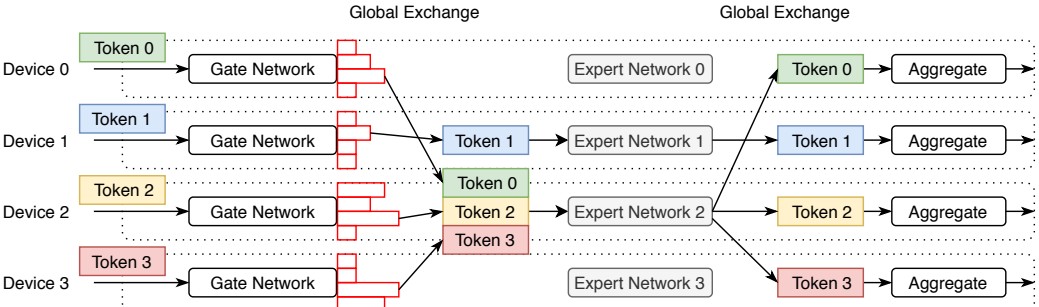

Figure 1: The popular expert parallelism method of MoE.

A MoE layer consists of a gate network $G$ and a set of N expert networks $\mathcal{E}_0, \ldots, \mathcal{E}_{N-1}$. For the gate module, the softmax activation function is widely used, reflecting each expert's normalized fitness for dealing with an incoming sample. Usually, only the experts with the top $k$ fit scores are selected to process the sample. The final output $y$ of the MoE layer is the aggregation of computed results.

Expert parallelism has been one of the most popular methods in existing MoE training systems [21, 8]. As shown in figure 1, the $N$ experts are evenly assigned to $P$ devices, with each device $i$ holding $E = N/P$ experts $\mathcal{E}_{i*E}, \ldots, \mathcal{E}_{(i+1)*E-1}$. Besides, the input tokens are also evenly partitioned over multiple devices with different small batches of the same size $S$ in a traditional data-parallel way. For each process, the shape of the dispatched data is $[k * S, d]$, where $d$ represents the hidden size. Each expert receives a combined batch of tokens (Global Exchange) and then carries out the computation. During the global communication, the number of samples sent to $\mathcal{E}_e$ from process $i$ is $c_{ie}$, and the shape of the transferred samples is $[c_{ie}, d]$. Afterward, the expert sends the calculated result back with a similar global exchange pattern.

However, the number of the tokens processed by different experts may be highly imbalanced that a small group of experts may receive the majority of data, like Expert 2 in Figure 1. Therefore, a load-balance auxiliary loss term $l_{aux}$ [26] is added to the train loss as an overall loss function:

$$m_i = \sum_x G(x)/S, \quad l_{aux} = \sum_{e=0}^{N-1} (m_{ie} * (c_{ie}/S)) \tag{1}$$

The auxiliary loss can dynamically adjust the value of $c_{ie}$ into target $k * S/N$. To further ensure load balance, a uniform data process capacity $C$ is set for each expert in many MoE training systems. DeepSpeed-MoE [21] decomposes the expert capacity $C$ evenly into the local capacity for each process and prunes the exchange size by the local capacity directly: $c_{ie} \leq C_{ie} = C/P$. FastMoE [8] efficiently uses the capacity with two extra all-to-all communication for exchange sizes: $\sum_{i=0}^{P-1} c_{ie} \leq C$.

### 3.2 Network Topology

The network environments are very complicated for distributed training on modern GPU clusters. As shown in Figure 2, there are four kinds of typical network topologies: homogeneous, ring, symmetric tree and asymmetrical tree. Homogeneous and ring structures are frequently used topologies for the intra-node environment. For a homogeneous structure, devices are always connected by the network

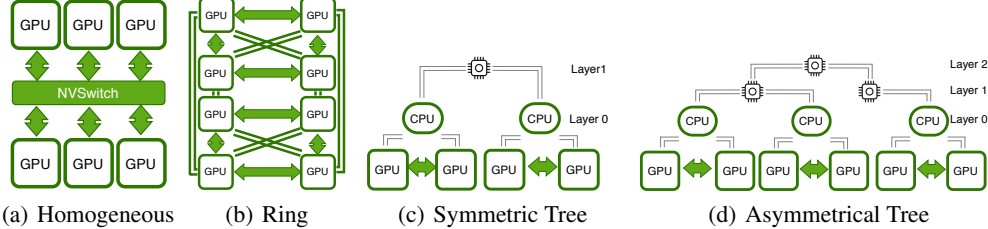

| (a) Homogeneous | (b) Ring | (c) Symmetric Tree | (d) Asymmetrical Tree |

Figure 2: Typical network topologies on modern GPU clusters. (a) A homogeneous node connected with NVSwitch. (b) A typical ring topology connected with NVLinks [19]. (c) A 2-layer symmetric tree topology of [2,2]. (d) A 3-layer asymmetrical tree topology of [[2,2],[2]].

with the same bandwidth, e.g., NVSwitch [20]. As for the ring topology, it is usually symmetrical. The bandwidths between adjacent devices may differ due to different numbers of connected links. The communication of nonadjacent devices has to hop through intermediate devices and the slowest link may become the bottleneck. Hierarchical tree is a common topology abstraction for multi-node distributed environments. Compared with the intra-node environment, inter-node links suffer from limited and volatile bandwidth (4∼25GB/s) and potentially degrade the communication performance. For convenience, we denote a tree topology as a nested list where the elements within the same sub-list are connected by the same switch. For a symmetric tree structure, we use $L_i$ to represent the number of the child nodes of each node in layer $i$. As for an asymmetrical tree structure, it is the most common topology for distributed training, which can be very irregular.

### 3.3 Motivation

The existing load-balanced data distribution of Equation 1 is unable fully exploit the complicated distributed environments. To demonstrate it, we set up an experiment on a [2, 2] symmetric tree topology cluster, where the devices are named $0$, $1$ (same node) and $\hat{0}$, $\hat{1}$ (same node), respectively. We dispatch 128MB [4] data with two dispatch patterns: (1) even dispatch and (2) uneven dispatch that a greater proportion of data is exchanged with a neighbor device. Table 1 shows the detailed dispatch proportions and the corresponding performance. Compared with even dispatch, uneven dispatch improves the overall communication performance by roughly 30%. This is mainly because the communication stress on inter-node links is relieved by transferring a smaller proportion of data. With the variety of distributed network topologies and their continuous development, the existing static even dispatch pattern is not effective enough. There is an urgent need for a routing strategy that can dynamically adapt to the underlying network environments.

Table 1: The communication performance of [[0,1],[$\hat{0}$, $\hat{1}$]] network topology.

|  | Dispatch Pattern | $0 \leftrightarrow 0$ | $0 \leftrightarrow 1$ | $0 \leftrightarrow \hat{0}$ | $0 \leftrightarrow \hat{1}$ | All |
|---|---|---|---|---|---|---|
| Ratio of data | Even | 1/4 | 1/4 | 1/4 | 1/4 | 1 |
|  | Uneven | 1/4 | 1/2 | 1/8 | 1/8 | 1 |
| Time ($\mu s$) | Even | 144 | 758 | 5609 | 5618 | 14019 |
|  | Uneven | 144 | 1492 | 2835 | 2861 | 10765 |

## 4 Topology Aware Routing Strategy

In this section, we first abstract the data dispatch problem into an optimization objective based on the communication model. Through some analysis, we obtain the target dispatch pattern under different topologies, which can eliminate the communication bottleneck during MoE training. Guided by the target pattern, we design a topology-aware routing loss to adaptively adjust the dispatch volume.

### 4.1 Communication Model

We characterize the communication cost using the well-known $\alpha$-$\beta$ cost model, where $\alpha$ and $\beta$ represent the fixed communication latency and the inverse bandwidth (i.e., transferring costs of each word), respectively. For convenience, $\alpha_{ij}$ and $\beta_{ij}$ are used to denote the latency and inverse bandwidth between the $i$-th and $j$-th GPU. During the training of MoE, the amount of data transferred

---

[4]Here, 128MB is used as a demonstration, which is the upper-bound transfer size of most of the typical MoE training tasks.

from GPU $i$ to $\mathcal{E}_e$ in GPU $j$ is $c_{ie} * d * b$, where $d * b$ is the transferred element size. To reduce the overheads of multiple send-receives between two GPUs, we merge the multiple small data chunks into a larger data chunk for delivery. The total amount of data delivered from GPU $i$ to GPU $j$ is $\sum_{e=E*j}^{E*(j+1)-1} c_{ie} * d * b$. A global data exchange consists of $P * P$ peer-to-peer data deliveries, among which the slowest delivery, as a lower-bound, constrains the final communication performance. Most of the global exchange implementations [21, 18, 24] are designed to approach the lower-bound. Therefore, our ultimate objective function is to minimize the slowest send-receive communication cost:

$$\min_c \max_{i,j}(\alpha_{ij} + \beta_{ij} * \sum_{e=E*j}^{E*(j+1)-1} c_{ie} * d * b) \tag{2}$$

For efficient MoE training, two constraint conditions should be satisfied. First, for any process $i$, the sent data size, i.e., $k * S$, should be equal to the sum of received data size of all experts:

$$k * S = \sum_{e \in \{0,...,N-1\}} c_{ie}, \forall i \in \{0, ..., P-1\}. \tag{3}$$

Second, to make full use of all the experts and pursue a better model accuracy, the data chunks dispatched to each expert should be balanced:

$$\frac{k * S}{E} = \sum_{i \in \{0,...,P-1\}} c_{ie}, \forall e \in \{0, ..., N-1\}. \tag{4}$$

## 4.2 Model Optimization

To get the target dispatch pattern, we need to solve the optimization problem in Equation 2. Nevertheless, Equation 2 contains plentiful parameters of a specific network, which complicates the solving process. Meanwhile, in some irregular topologies, some devices may suffer from quite limited bandwidth when communicating with other devices. According to Equation 2, the experts assigned to these devices may receive a quite small dispatch chunk size from the other processes, which may make the experts lack of sufficient data exchanges and lead to expert isolation phenomenon. To tackle these problems, we simplify the optimization problem to accelerate the solving process and smooth the values of $\alpha_{ij}, \beta_{ij}$ for an approximate result to prevent expert isolation. Since each send-receive communication shares the same $\alpha, \beta$ in homogeneous network, the target dispatch chunk size $\hat{c_{ie}}$ is equal to the load-balanced chunk size $\frac{k*S}{N}$. In the following part, we focus on the analysis of the optimization problem under heterogeneous topologies.

On a $n$-layer symmetric tree topology, for any device $i$, all the devices can be split into $n$ sub-groups of $G^i = \{G^i_t | t < n\}$. $G^i_t$ is the group of devices whose shortest path from device $i$ are across $t$ switches. Multiple hops in cross-switch communication will suffer from extra overheads and the most limited bandwidth in the hops dominates the final bandwidth. Therefore, we can simplify the original $\alpha_{ij}, \beta_{ij}$ into $n$ value: $\alpha_l = \frac{\sum_{i<j} \mathbb{I}(j \in G^i_l) * \alpha_{ij}}{(\prod_{k=0}^l L_k) * (L_l-1)/2}, \beta_l = \frac{\sum_{i<j} \mathbb{I}(j \in G^i_l) * \beta_{ij}}{(\prod_{k=0}^l L_k) * (L_l-1)/2}$, which can precisely characterize the underlying topology and eliminate the noise of profiling. Then we get a hierarchical matrix $\hat{\alpha}, \hat{\beta}$:

$$\hat{\alpha}_{ij} = \sum_l \mathbb{I}(j \in G^i_l) * \alpha_l, \hat{\beta}_{ij} = \sum_l \mathbb{I}(j \in G^i_l) * \beta_l \tag{5}$$

Take equation 3 and 4 as optimization constraint conditions, we simplify the optimization problem as follows:

$$\min_c T^{lower}_{comm} = \min_c \max_{i,j}(\hat{\alpha_{ij}} + \hat{\beta}_{ij} * \sum_{e=E*j}^{E*(j+1)} c_{ie} * d * b)$$

$$s.t. \quad \frac{k * S}{E} = \sum_{i \in \{0,...,P-1\}} c_{ie} * d * b, \forall e \in \{0, ..., N-1\},$$

$$k * S = \sum_{e \in \{0,...,N-1\}} c_{ie} * d * b, \forall i \in \{0, ..., P-1\}, \tag{6}$$

$$c \geq 0$$

Then, we can get the near optimal solution after omitting the small latency term:

$$\hat{c}_{ie} = \frac{k * S}{E * \sum_j \frac{1}{\hat{\beta}_{ij}} * \hat{\beta}_{i\lfloor \frac{e}{E} \rfloor}} \tag{7}$$

The optimal data distribution of the above min-max problem is only related to the bandwidth: the volume of $\hat{c}_{ie}$ is linear to the bandwidth. The rationale behind the optimal result is that higher bandwidth links should bear more loads for an overall communication balanced workflow. The ring topology also shows a hierarchical characteristic and the solution for ring topology has the same pattern as symmetric trees.

Under some irregular asymmetric topology, the optimal result of Equation 2 may result in some experts assigned to the devices of most limited bandwidth in lack of data exchanges with the rest of the devices. Compared to the data distribution of other experts, data chunks from local devices occupy a larger proportion of received data chunks in those isolated experts. For the fairness among the experts, we transform the asymmetric topology into a symmetric one by merging the separate nodes into the close symmetric sub-trees. For example, [[2,2][2]] in figure 2(d) can be merged as symmetric structure [[2,2,2]]. After that, we can optimize the lower bound of communication as the symmetric structure.

### 4.3 Routing Strategy

Once getting the target data dispatch volumes among processes, we can use it to guide the MoE training. To not sacrifice the model accuracy, instead of setting a compulsory dispatch ratio directly, we design a topology-aware adaptive routing loss.

$$p_i = Norm(1/\hat{c}_i), \quad l_{topo}^i = N * P * \sum_{e=0}^{N-1} (p_{ie} * m_{ie} * (c_{ie})/S) \quad \forall i \in \{0, ..., P-1\} \tag{8}$$

As shown in Equation 8, we set a penalty weight $p_i$ as the adjustment coefficients for the topology loss $l_{topo}^i$ of each process $i$. We set the normalized $1/\hat{c}_i$ as $p_i$ to make sure $c_{ie}$ approximates the value of $\hat{c}_{ie}$. Normalization functions that enlarge the penalty of the low-bandwidth transfer, e.g., softmax, are also preferable. As shown in the calculation of $l_{topo}^i$, the data dispatched to $\mathcal{E}_e$ with limited bandwidth will suffer from a larger penalty weight $p_{ie}$. Despite of these penalty modifications, auxiliary loss occupies a small proportion of the final loss, and the value of the auxiliary loss decreases when experts' number scales up. Therefore, our final topology loss is expanded $N * P$ times to keep the magnitude of loss value.

There are a number of advantages of the proposed topology-aware strategy. On the one hand, compared to setting a compulsory dispatch ratio, the proposed loss can adjust the communication volume to fit in the underlying topologies in a mild way without damaging the convergence. A compulsory dispatch ratio has a high potential to overwhelm the influence of the train loss and sacrifice the model accuracy. With the topology loss, the train loss can still dominate in the final loss value. As a result, the dispatch results are mainly influenced by the train loss for a better model accuracy. Besides, the topology-aware strategy has more potential to utilize the token information for efficient sparse training. Guided by the topology-aware loss, the tokens nearby are more likely to be processed by the same expert. Since the correlation of adjacent tokens contains the vital information in sparse attention [10], the experts in sparsely gated routing structure may also be more likely to extract important information from adjacent tokens.

In addition, the topology-aware loss can be easily incorporated to existing MoE systems, such as DeepSpeed-MoE and FastMoE. Taking FastMoE as an example, one just needs to directly replace the popular load-balanced loss $l_{aux}$ with the proposed topology-aware loss $l_{topo}$. Since local capacity threshold $C_{ie}$ is adopted in DeepSpeed-MoE for load balance, one can modify the local capacity sizes to be consistent with the proposed dispatch pattern by setting $C_{ie}$ to be proportional to the target data chunk sizes $\hat{c}_{ie}$. Instead of padding the data chunks with extra zeros to be the same size as DeepSpeed-MoE, one all-to-all communication is added to get the information of send-receive data chunk sizes and dispatch data chunks according to the sizes.

## 5 Evaluation

**Experiment setup**    To demonstrate the effectiveness of TA-MoE, we carry out a series of experiments on three typical NVIDIA GPU clusters with different network topology, and some representative model configurations. Table 2 lists the cluster settings. The testbed is three typical clusters from the PaddleCloud[5] platform. For cluster A, each node consists of 8 NVIDIA Tesla 40GB A100 GPUs connected with NVSwitch, which shows high performance for both computation and network communication. Clusters B and C are equipped with 8 NVIDIA Tesla 32GB V100 GPUs in each node. The nodes in cluster B are connected by the same switch, while cluster C is composed of a large number of servers and switches that are interconnected through an internal network infrastructure. Besides, the software configurations are set as CUDA 11.0, NCCL 2.8.4 and CUDA 11.1, NCCL 2.8.3, for cluster A and cluster B, C, respectively.

Table 2: Cluster Setting.

| Clusters | GPU | Intra-Node | Inter-Node | Symmetric | Same switch |
|---|---|---|---|---|---|
| A | 40G-A100 | NVSwitch | 100GB/s RoCE/4 | x | x |
| B | 32G-V100 | NVlink | 100GB/s RoCE/8 | ✓ | ✓ |
| C | 32G-V100 | NVlink | 100GB/s RoCE/8 | x | x |

Since the transformer architecture is the most common base structure of MoE, we focus the experiments on problems related to transformer-based model, with GPT-3 Medium [3] as the base model and multi-layer perception as the expert. In our experiments, the number of the experts are chosen among {8, 16, 32, 48, 64} with each device deployed with one expert. Both the Switch top-1 [7] and the GShard top-2 gates [11] are tested with the weight of auxiliary loss set as 1.0. For the consistency of the experiment, we implement the models by a single framework Paddle [2] and train on the open-source openwebtext2 dataset [1]. More detailed specifications of model settings can be found in Table 3. To be more general, we also add the tests of MoE on Swin Transformer based MoE tasks in Section A.3 of the Appendix. These results further demonstrate the effectiveness of the proposed topology-aware routing algorithm on different model architectures.

Table 3: Detailed specifications of the GPT models.

| Gate | Layers | Hidden size | Intermediate size | Batch size | Data type | Capacity factor | Clusters |
|---|---|---|---|---|---|---|---|
| Switch | 12 | 1024 | 4096 | 6 | FP16 | 1.0 | A |
| GShard | 12 | 1024 | 2048 | 6 | FP16 | 2.0 | A |
| Switch & GShard | 12 | 1024 | 2048 | 4 | FP32 | 1.2 | B |
| Switch & GShard | 12 | 1024 | 2048 | 4 | FP32 | 1.2 | C |

**Methodology**    We incorporate TA-MoE into widely used DeepSpeed-MoE [21] and FastMoE [8] implementations. Because TA-MoE modifies the gate structure, we first compare the validation loss w.r.t. steps to ensure that TA-MoE will not interfere with the convergence on various model scales. On top of that, we test the overall throughput and the speedup of TA-MoE over these two classical baselines. To be more comprehensive, we also compare with the recently proposed FasterMoE Hir gate [9] on the metric of time to convergence performance. Besides, a detailed analysis of communication costs, as well as the distribution of the dispatch are also given.

**Accuracy and Performance**    Like some well-recognized related works, e.g., BASE layer and DeepSpeed-MoE, we have depicted the validation performance of every fixed interval step in Figure 3 as the comparison metric. We first compare the validation loss w.r.t steps of TA-MoE and the representative FastMoE on cluster C. As shown in Figure 3, the loss curves of TA-MoE and FastMoE are consistent to converge under different training scales of 8 experts to 48 experts. These results demonstrate that

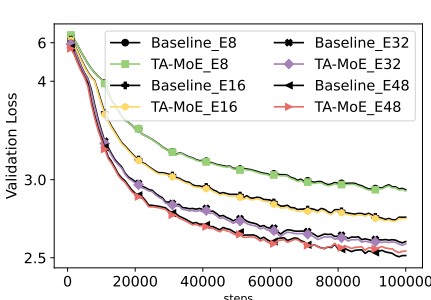

Figure 3: Validation loss w.r.t. steps.

---

[5]A Cloud Platform of Baidu Inc.

the TA-MoE can adaptively adjust the data dispatch volume without sacrificing the model accuracy. To be more comprehensive, we also add the perplexity (PPL) metric in Table 4 of the Appendix, which gives similar effective results.

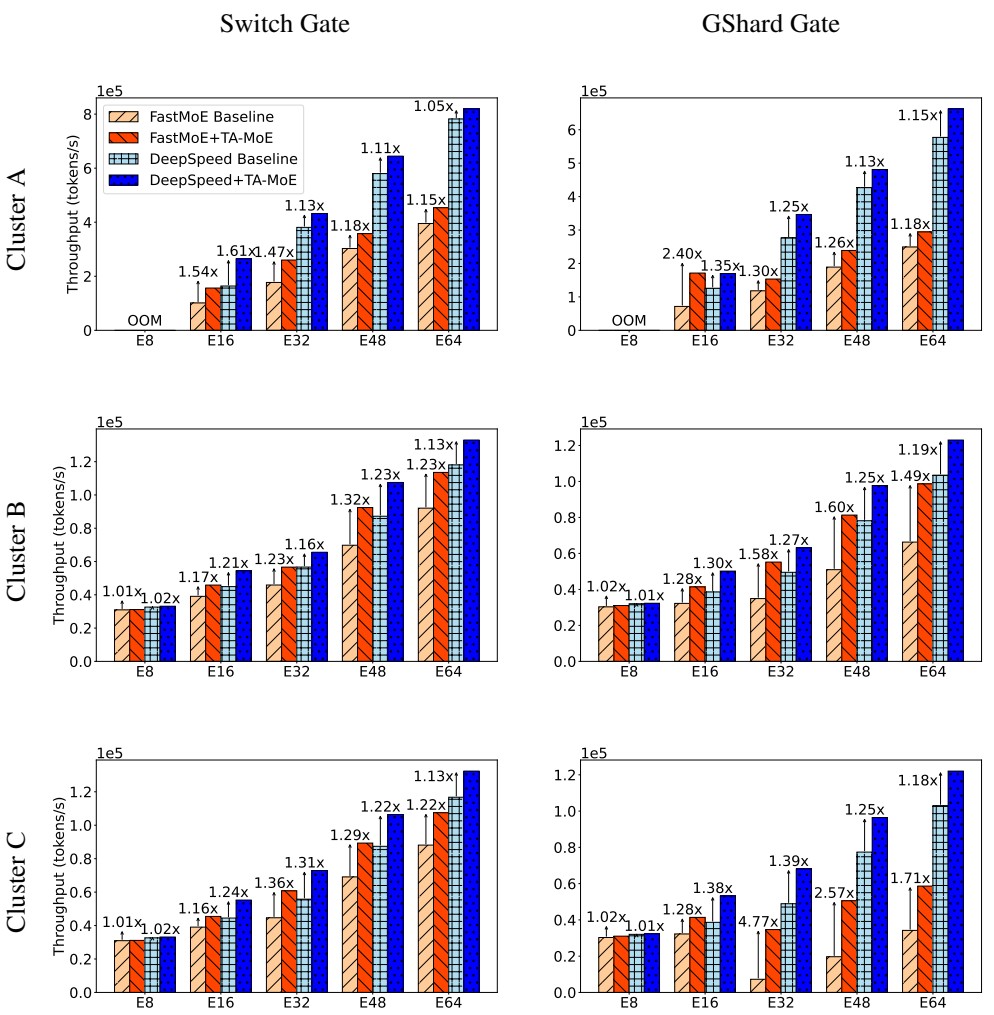

Figure 4: Performance of TA-MoE over DeepSpeed-MoE and FastMoE under different hardware and model settings.

To demonstrate the advantages of TA-MoE in terms of training performance, we compare it with both DeepSpeed-MoE and FastMoE on various hardware and model configurations. The performance indicators including throughout (tokens/s) and the speedups are depicted in Figure 4. It is clear that TA-MoE can bring significant performance improvements over its counterparts under almost all the configurations. When compared with the DeepSpee-MoE, the achieved speedup is about 1.05x-1.61x. As for FastMoE, the performance improvement is around 1.01x-4.77x. More speedups are achieved on Fast-MoE than DeepSpeed-MoE. This is because TA-MoE has a more dynamic dispatch pattern and larger adjustable space based on FastMoE.

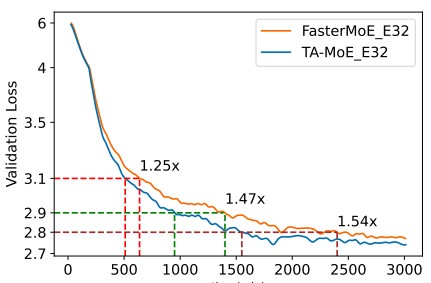

Figure 5: Comparison with FasterMoE.

It is also observed that more improvements are obtained on Cluster C, which can reach 4.77x for some cases, due to the relief of its serious network contention of cross-switch communication. In addition, the comparison of the results of Switch and GShard gate reveals that TA-MoE can behave better in adjusting larger volume of data.

To be more comprehensive, we make further comparisons with the recently proposed FasterMoE [9]. Because the compulsory dispatch strategy of FasterMoE affects the convergence, we take the validation loss w.r.t time as the comparison metric. Clusters C are selected as the representative testing clusters. As shown in Figure 5, TA-MoE can converge faster than FasterMoE. We evaluate the time to reach the validation loss values of 3.1, 2.9, and 2.8, and TA-MoE can converge faster by about 1.25x, 1.47x and 1.54x. The results further verify that the proposed adaptive routing loss is more effective than the compulsive dispatch method, which sacrifices model accuracy for training performance.

**Communication Analysis** To better show the effects of the dynamical dispatch strategy, we further analyze communication and computation cost, and the distribution of data dispatch on cluster C. As shown in Figure 6(a), thanks to the proposed TA-MoE strategy, the communication cost is reduced rapidly, with roughly 1.16x to 6.4x speedups. It is also observed that the maximum speedup is achieved for 32 experts on four cross-switch nodes. This is because the four nodes are deployed under four different switches, and cross-switch links severely bottleneck the data exchange. Once the tension is relieved, the obtained benefits can be dramatic. In addition, we visualize the dispatch pattern of an example with 64 expert by depicting the number of the tokens of Rank 0-7 sending to other ranks. More details of the dispatch distributions for other expert scales are attached to Section A.2 of the Appendix. In Figure 6(b), as expected, most of the data of Rank 0-7 are dispatched to low-overheads nearby ranks, which further verifies the effectiveness of adaptive topology-aware loss.

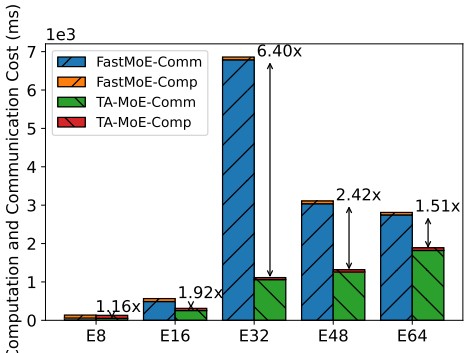
(a) Breakdown of communication and computation.

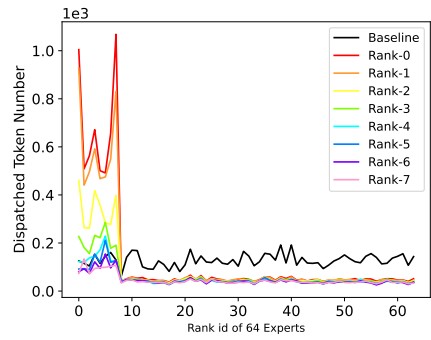
(b) Distribution of data dispatch of Rank 0-7.

Figure 6: Analysis of communication and computation cost and the distribution of data dispatch.

# 6   Conclusion

In this paper, a topology-aware routing strategy, TA-MoE, was proposed to stress the mismatch between the data dispatch pattern and the network topology. Based on communication modeling, we abstract the dispatch problem into an optimization objective and obtain the approximate dispatch pattern under different topologies. On top of that, a topology-aware auxiliary loss was designed, which can adaptively route the data to fit in the underlying topology without sacrificing the model accuracy. Experiments show that the proposed method can substantially outperform its counterparts on a variety of the hardware and model configurations, with roughly 1.01x-1.61x, 1.01x-4.77x, 1.25x-1.54x improvements over the popular DeepSpeed-MoE, FastMoE and FasterMoE systems. In the future, we plan to take more delicate communication operators into consideration of dynamic dispatch pattern and extend our work to more hardware environments.

# 7   Acknowledgement

This work was supported in part by National Key R&D Program of China (2021ZD0110501).

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

# A  Appendix

## A.1  Perplexity Evaluation Results

Table 4: **Perplexity Evaluation Result.** To further validate the model convergence performance, we list the perplexity (PPL) at 10w step (near 7 days) of GPT-Medium (12 layers, hidden size 1024, intermediate size 2048, GShard, Capacity factor 1.2) with different expert numbers on the openwebtext2 dataset. Combined with the information in Figure 3 of the paper, we can find that TA-MoE does not influence the convergence of the model when compared with the well-known FastMoE baseline.

| Expert Scale | TA-MoE Valid PPL | Baseline Valid PPL |
|---|---|---|
| 8 | 17.97 | 18.12 |
| 16 | 15.18 | 15.39 |
| 32 | 13.37 | 13.53 |
| 48 | 12.55 | 12.49 |

## A.2  Data Dispatch Distribution

Figure 7 further elaborates the data dispatch patterns of GPU rank 0-7 (within a node) on 8, 16, 32, 48 experts. We also list the Rank-0 dispatch information of FastMoE with even distribution method as the baseline. On a single node topology, each rank prefers to dispatch data to the expert within a node. The topology influence on the data dispatch preference is relatively small, because the bandwidth variance within a node is small. Multi-node topology results show a consistent "ladder-like" distribution trend that the ranks within a node has a high preference to dispatch the data to intra-node rank group, instead of transferring data to inter-node ranks. These results verify the effectiveness of the proposed adaptive topology-aware method.

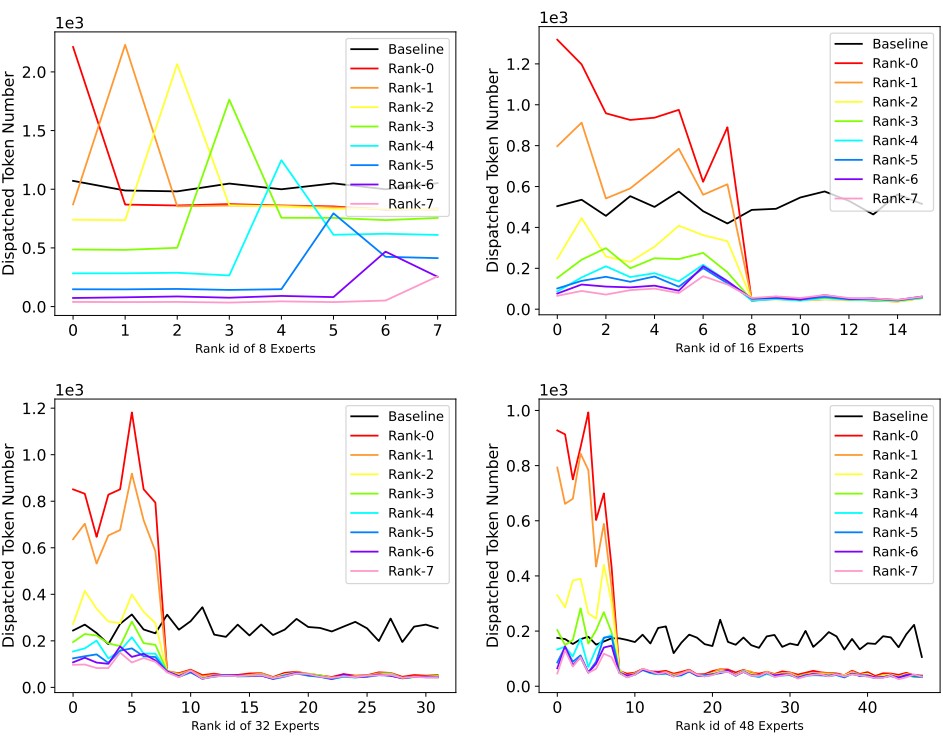

Figure 7: The data dispatch distribution of rank 0 to 7 on 16, 32, 48 GPUs and experts.

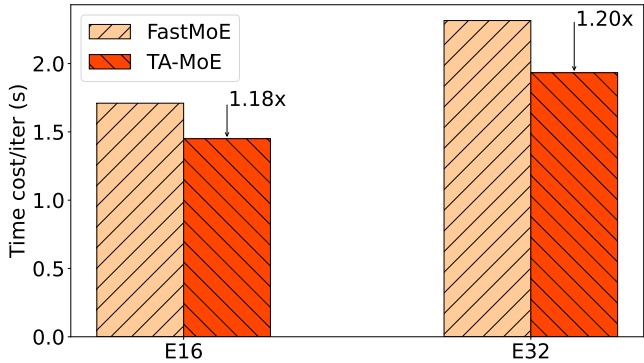

Figure 8: Speedup of TA-MoE over FastMoE on Swin Transformer Based Model.

### A.3 Tests on Swin Tranformer Based MoE Tasks

To further validate the generality of TA-MoE, we also carry out an experiment of Vision Tasks on ImageNet-1k dataset. The well-known vision transformer architecture Swin Transformer [14] is picked as the base model. The detailed model configurations are listed in Table 5. We evaluate the speedup on Cluster A with 16 and 32 GPUs as an illustration, where 16 GPUs configurations represents symmetric tree topology and 32 GPUs configurations represents asymmetric tree topology. As shown in Figure 8, we can achieve 1.18x and 1.20x speedup when compared with FastMoE on 16, and 32 GPUs, respectively.

Table 5: Detailed specifications of the Swin-Transformer model.

| Name | Layers | Gate | Stage 1 | Stage 2 | Stage 3 | Stage 4 | Capacity factor |
|---|---|---|---|---|---|---|---|
| Swin Transformer v1 | 12 | GShard | concat 4x4, 96-d, LN {win.sz. 7x7, dim 96, head3} x2 | concat 2x2, 192-d, LN {win.sz. 7x7, dim 192, head6} x2 | concat 2x2, 384-d, LN {win.sz. 7x7, dim 384, head12} x6 | concat 2x2, 768-d, LN {win.sz. 7x7, dim 768, head324} x2 | 1.2 |

