# OpenReview forum: "TA-MoE: Topology-Aware Large Scale Mixture-of-Expert Training"
_NeurIPS.cc/2022/Conference — NeurIPS 2022 Accept_

### Official Review · Reviewer_QoUG · 2022-07-10

**Rating:** 6
**Confidence:** 2
**Soundness:** 3 good
**Presentation:** 3 good
**Contribution:** 3 good

**Summary:**

Sparsely gated Mixture-of-Expert (MoE) plays a vital role in large-scale model training but suffers from both load imbalance and global communication. In addition, the existing even dispatch approach may cause network contention and worsen the previous challenges. This work proposed a topology-aware large-scale MoE training method, called TA-MoE, that can adapt communication volume to fit the underlying network topology without interfering with the model convergence. The key ideas are abstracting the dispatch problem as a communication cost optimization problem and then adding an auxiliary loss with pattern-related coefficients. Experiments show that TA-MoE provides up to 1.61x speedup and 4.77x speedup over DeepSpeed-MoE and FastMoE without accuracy loss.

**Questions:**

Please refer to the above weakness part.

**Limitations:**

The authors have adequately addressed the limitations and potential negative societal impact of their work.

**Strengths And Weaknesses:**

Strengths:
+ this work tried to tackle a very significant and interesting challenge in MoE system: network topology may worsen the communication and load balance problems during the dispatch in MoE.
+ the paper is well organized and easy to follow
+ the proposed TA-MoE method is simple and effective: extensive experiments show that TA-MoE is able to offer noticeable speedup over the state-of-the-art under different hardware  and model configurations.

Weaknesses:
- the experiments are mostly doen with GPT models; it would be better to have models with different neural architectures in the evaluation benchmark. It is unclear how TA-MoE works on other MoE using models other than GPTs.

---

> ### Author Response · Authors · 2022-08-02
> **Response for Review QoUG**
>
> Thanks for your valuable suggestion. Please find our response below.
>
> 1)  **About testing on more model architectures**
>
> To be more general, we have added the tests of MoE on Swin Tansformer based MoE tasks in section 1.3 of the Appendix. The results have further demonstrated the effectiveness of the proposed topology-aware routing algorithm on different model architectures.

---

### Official Review · Reviewer_VbEU · 2022-07-10

**Rating:** 5
**Confidence:** 5
**Soundness:** 3 good
**Presentation:** 2 fair
**Contribution:** 2 fair

**Summary:**

The paper proposes a new algorithm to improve training efficiency of Mixture of Experts models in a distributed training setting by exploiting the network topology information. To achieve this, the authors propose a new auxiliary loss term incorporating communication bandwidth to encourage tokens to be routed to closer nodes rather than further nodes. By applying this new algorithm, authors claim that they could achiever faster throughput (1.01x - 4.77x) without losing accuracy on their several different clusters. As a result, they show a faster wall-clock time convergence.

**Questions:**

There are a few questions that need to be answered as well. First, it is not clear that how capacity and overflow tokens are handled in the proposed algorithm. They are known to be critical factors for the successful MoE model trainings, but not much details about those are included in the paper.
Second, it is not clearly shown how the locality preferred routing impacts the training compared to the global routing. Maybe, it could be useful to have a few data points such as forced local routing. It is unclear how experts can have expertise while trained by local tokens more. It might be useful to see how different topology affects expertise of different experts.
Lastly, a few details are missing (number of layers, datasets, model architecture). Figure 3 and Figure 6 (b) are hard to read.

**Limitations:**

This paper is focusing on the computation algorithm itself. So, it might not have direct societal impact.

**Strengths And Weaknesses:**

The communication overhead is one of the major issues for the MoE model training and this paper proposes a new method to deal with this problem naturally. Given the increased usage of MoE model technology, this is a timely work. Having a soft guidance seems like a good idea not to hurt the original training dynamics while encouraging locality of token routing. And, as authors mentioned, there have not been this kind of topology aware loss terms before as far as I know.
However, there are a few missing details about model configurations and algorithms asked in the question section. And, the overall speed gain is minor.

---

> ### Author Response · Authors · 2022-08-02
> **Response to reviewer VbEU**
>
> Thanks for your valuable feedbacks. Our responses are listed below.
>
> 1)  **About the capacity controlling method**
>
> We consider two different capacity controlling methods used in DeepSpeed-MoE and FastMoE (line 109-112). The details on how to handle the capacity based on these two implementations have been introduced in line 221-228. For FastMoE, we only need to replace the original balance loss with topology loss and the capacity controlling method remains the same. As for DeepSpeed-MoE, every data chunk $c_{ie}$ from process $i$ to expert $e$ is pruned by $\hat{c_{ie}}$, an unevenly split part of $C$. Section 4.2 shows how to get the optimal sub capacity size $\hat{c_{ie}}$ based on topologies.
>
> 2) **About the locality routing algorithm**
>
> Compared with global routing and forced routing, TA-MoE is advantageous in the following aspects:
>
> TA-MoE can take advantage of the adjacent information of data, which can be revealed in the dispatch patterns in Figure 6(b) and section 1.2 of Appendix. The figures reveal a “ladder-like” trend that the ranks within a node have higher dispatch preference to intra-node rank groups. As a result, experts prefer to receive “adjacent tokens” from nearby ranks. In fact, the correlation of adjacent tokens usually contains more important information, which makes TA-MoE be potential to maintain the accuracy and achieve higher performance.
>
>  TA-MoE has some advantages over existing compulsive local routing works, e.g., FasterMoE (line 85-87). Firstly, TA-MoE achieves better time to convergence performance as shown in Figure 5 and line 279-282. Secondly, a forced local routing needs a manually set proportion of local data chunk size, which makes it hard to adapt to various topologies. Lastly, compulsive local routing usually requires extra local sorting operations, introducing computation overheads.
>
>  3)  **About GPT model configurations**
>
> The information of the GPT model, such as the dataset and model architecture, have been introduced in the second paragraph of section 5. We add the number of layers in Table 2 of the paper.
>
>  4)  **About Figures**
>
> Figure 3 and Figure 6 (b) have been improved in the revised paper.

---

> > ### Comment · Reviewer_VbEU · 2022-08-08
> > **Thanks for the response.**
> >
> > Thank you for the clarifications on my questions.
> > Based on those, I would increase my rating to 5.
> >
> > Also, I have one follow-up question regarding Figure 3.
> > It seems there are data points that TA-MoE gives worse loss value when the number of experts increases (at the same number of updates).
> > Do you have any experiments with more experts or more updates with E48 case?

---

> > > ### Author Response · Authors · 2022-08-09
> > > **Response to the second feedback.**
> > >
> > > We appreciate the constructive feedback and valuable suggestions again.
> > >
> > > We apologize that we are currently not able to have experiments with more experts or more updates due to the limitations of the computation resources. Please kindly notice that in the revised Figure 3, we have enlarged the value scale from 2.5-3.0 to be clearer. As a result, the fluctuation of curves combined with magnified scale might amplify the impression of loss difference. In fact, the loss curves of TA-MoE are consistent to the baselines with a few fluctuations, which we think is acceptable. To further demonstrate the consistency, we also attached the PPL metric result in Table 1 of the supplementary materials. Under configurations with 8, 16, 32 and 48 experts, the PPL differences, i.e., baseline valid PPL minus TA-MoE valid PPL, are +0.15, +0.21, +0.16 and -0.06. Although with 48 experts the PPL value (the less, the better) is a little higher than the baseline, the extent (0.06) is much lower than the others (0.15, 0.21 and 0.16), where TA-MoE shows better PPL results. Therefore, we can conclude that the difference is an acceptable fluctuation.

---

> > > > ### Comment · Reviewer_VbEU · 2022-08-09
> > > > **Thanks for the clarification.**
> > > >
> > > > I agree with your point. I will keep my current rating 5 considering all of those. Thanks!

---

### Official Review · Reviewer_EndF · 2022-07-11

**Rating:** 5
**Confidence:** 3
**Soundness:** 3 good
**Presentation:** 3 good
**Contribution:** 3 good

**Summary:**

This paper addresses the problem of MoE routing under the cases of different network topologies by allocating another abstraction layer for the topology and designing an auxiliary objective to optimize. Experiments show very good improvement in terms of speed compared to strong baselines.

**Questions:**

As said, while many (if not most, in my humble opinion) AI people are not working near the system level to address the same problems as in this paper, e.g., they simply use AWS, Azure, Colab, or other cloud infrastructure for fast prototyping or deployment, I wonder how much improvement this solution can make with different cloud platforms? In other words, have “cloud” people already seen those heterogeneous infrastructure problems and addressed that at the cloud provision/hypervisors level? The paper presents the case of PaddleCloud but how about others?

**Ethics Review Area:**

["I don’t know"]

**Limitations:**

Maybe not very relevant since the paper addresses the system-related level and thus is hard to judge those impacts.

**Strengths And Weaknesses:**

Strength:

1. The paper offers an important contribution to the AI community at the system level, which is probably not difficult to approach for many people working in this field. In fact, in my humble opinion, not so many AI people have the opportunity to access detailed hardware information as cloud users such as with Azure or AWS.

2. The experiments show very good improvement over strong baselines. System analysis is clearly presented.

Weakness

1. The paper addresses the system level. However, since it claims a significant boost of speed without sacrificing the model accuracy, it needs to show the accuracy, e.g. at least the LM-related one with NLP-related metrics.

2. Line 240, which claims "without loss of generality", is probably too strong. My suggestion is if the solution is good, with the current hardware settings, the authors can run current codes for other many applications of which codes are available to further solidify their claims.

3. Likewise, why not show the data dispatch distribution of other ranks but only rank 0? If space is limited, appendix space is always there.

4. In the era of GPUs and large data, the motivation is led by demonstrating only 128MB of data is probably inefficient. Probably at least some GBs, or even stronger in a combination with different types of data would make a stronger motivation.

5. No code is provided.

---

> ### Author Response · Authors · 2022-08-02
> **Response to reviewer EndF**
>
> Thank you for giving inspiring suggestions and pointing out insufficient details. Our responses are listed below.
>
> 1) **About the accuracy metric**
>
> Like some well-recognized related works, e.g., BASE layer and DeepSpeed-MoE, we have depicted the validation performance of every fixed interval step in Figure 3 as the comparison metric. To be more comprehensive, we have added the perplexity (PPL) metric in Table 1 of the Appendix.
>
> 2) **About testing on more model architectures**
>
> To be more general, we have added the tests of MoE on Swin Transformer based MoE tasks in section 1.3 of the Appendix. The results have further demonstrated the effectiveness of the proposed topology-aware routing algorithm on different model architectures.
>
> 3) **About data dispatch distribution of other ranks**
>
> The data dispatch of all the ranks have been shown in section 1.2 of the Appendix, which shows similar conclusions as rank 0.
>
> 4) **About the transfer volumes in the motivation**
>
> In the motivation section, 128M data is used as an example to show that the static load-balanced dispatch is not effective on complex distributed environments. In fact, the value of the load-balanced transfer volumes has little effects on the demonstrated phenomenon. To be more representative, we choose 128M as an illustration, which is close to the real chunk size in most of the MoE training experiments.
>
> 5) **About the code**
>
> We will publish our code after the paper is accepted as soon as possible.
>
> 6) **Discussions on cloud platforms**
>
> The cloud provisions/hypervisors concentrate more on high-level schedules of different tasks. We focus on the single MoE task itself. They are orthogonal to each other. We observed that the dynamic feature and global data exchange pattern of MoE training mismatch the hierarchically heterogeneity of distributed cloud environments. To solve it, we propose a topology-aware data dispatch algorithm. In fact, cloud architectures share most of the topology abstractions analyzed in our paper. We therefore believe our algorithm can achieve similar performance improvements when it is applied to other cloud platforms.

---

> > ### Comment · Reviewer_EndF · 2022-08-08
> > **Response to First Response**
> >
> > I thank the authors for their efforts in addressing many of my questions and concerns and other reviewers' as well. I hope the rest will be answered in later versions of the paper and I am retaining my score of borderline accept for now.

---

> > > ### Author Response · Authors · 2022-08-09
> > > **Response to the feedbacks**
> > >
> > > Thank you for the valuable feedback. We were not aware that there are questions left unanswered. Could you kindly let us know which questions require further clarifications? Thank you again.

---

### Meta-Review · Area_Chair_Usnn · 2022-08-26

**Recommendation:** Accept
**Confidence:** Less certain

**Metareview:**

Mixture-of-Expert (MoE) models have demonstrated a lot of success recently. To further improve upon the existing literature this paper studies MoE routing for different network topologies. This is essentially to deal with the communication overhead of MoE training. The strategy is to add another layer on top for the topology along with a corresponding objective to optimize. The authors also provide experiments demonstrating improved speed of convergence. The reviewers were in general positive and liked the idea of the paper. The reviewers did however raise issues about lack of clear demonstration that accuracy is not compromised, lack of large data, and a few other more technical concerns. The reviewers concerns seem to be more or less addressed by the authors. My overall assessment of the paper is positive. I think the general premise of the paper is interesting and the paper has interesting ideas. I do agree however that the experiments need to be more thorough. I am recommending acceptance but request that the authors follow the reviewers comments to improve their experimental results

**Award:**

No

---

### Decision · Program_Chairs · 2022-09-14

Accept